# Sex Differences in Response to Marek’s Disease: Mapping Quantitative Trait Loci Regions (QTLRs) to the Z Chromosome

**DOI:** 10.3390/genes14010020

**Published:** 2022-12-21

**Authors:** Ehud Lipkin, Jacqueline Smith, Morris Soller, David W. Burt, Janet E. Fulton

**Affiliations:** 1Department of Genetics, The Alexander Silberman Institute of Life Sciences, The Hebrew University of Jerusalem, Edmond J. Safra Campus, Givat Ram, Jerusalem 91904, Israel; 2The Roslin Institute and Royal (Dick) School of Veterinary Studies R(D)SVS, University of Edinburgh, Easter Bush, Midlothian EH25 9RG, UK; 3Hy-Line International, P.O. Box 310, 2583 240th St., Dallas Center, IA 50063, USA

**Keywords:** Marek’s Disease, QTLR, chicken Z chromosome, sexual dimorphism, linkage disequilibrium, LD blocks

## Abstract

Marek’s Disease (MD) has a significant impact on both the global poultry economy and animal welfare. The disease pathology can include neurological damage and tumour formation. Sexual dimorphism in immunity and known higher susceptibility of females to MD makes the chicken Z chromosome (GGZ) a particularly attractive target to study the chicken MD response. Previously, we used a Hy-Line F_6_ population from a full-sib advanced intercross line to map MD QTL regions (QTLRs) on all chicken autosomes. Here, we mapped MD QTLRs on GGZ in the previously utilized F_6_ population with individual genotypes and phenotypes, and in eight elite commercial egg production lines with daughter-tested sires and selective DNA pooling (SDP). Four MD QTLRs were found from each analysis. Some of these QTLRs overlap regions from previous reports. All QTLRs were tested by individuals from the same eight lines used in the SDP and genotyped with markers located within and around the QTLRs. All QTLRs were confirmed. The results exemplify the complexity of MD resistance in chickens and the complex distribution of *p*-values and Linkage Disequilibrium (LD) pattern and their effect on localization of the causative elements. Considering the fragments and interdigitated LD blocks while using LD to aid localization of causative elements, one must look beyond the non-significant markers, for possible distant markers and blocks in high LD with the significant block. The QTLRs found here may explain at least part of the gender differences in MD tolerance, and provide targets for mitigating the effects of MD.

## 1. Introduction

Sexual dimorphism in immunity is widely reported [1,2]. Although in some studies females were found to be more susceptible to some infections [3], in many vertebrates they are more immunocompetent than males [4]. Males exhibit higher susceptibility to infection [5], are less able to cope with infection [6], or suffer more severe symptoms than females [7].

In diploid species, sex chromosomes occur either as heterogametic females (ZW) and homogametic males (ZZ), or homogametic females (XX) and heterogametic males (XY) [8]. Degeneration of chromosomes W (avian) or Y (mammals) during the evolution of the sex chromosomes can result in only a single functional allele in the heterogametic sex, a dosage imbalance often not tolerated during development [9]. Consequently, dosage compensation (DC) evolved to balance expression between X or W and the autosomes on one side, and to balance X or W expression between the sexes on the other side.

DC is documented in many species, including both animals and plants. Nevertheless, many organisms do not display chromosome-wide expression equalization of sex chromosomes. It seems that chromosome-wide DC is much more frequent in XY systems, while it is comparably rare in ZW species [9]. Indeed, expression analysis did not support inactivation on the chicken Z Chromosome [8]. This accords well with the report that, in chicken, the Z and W chromosomes are almost completely differentiated [10]. The almost complete differentiation also implies a minimal size of the chicken pseudoautosomal region (PAR). Indeed, the sizes of pseudoautosomal regions tend to reduce from primitive groups of birds to more evolutionarily advanced [11]. The exact coordinates of the chicken PAR are yet to be completely defined, although various studies have attempted to map this region [11,12].

Marek’s Disease (MD) is responsible for an estimated ~2 billion USD annual loss to the global poultry industry through mortality, lost production and vaccination costs [13]. It is caused by the immunosuppressive Marek’s Disease Virus (MDV) [14]. MDV is a cell-associated oncogenic α herpesvirus which can cause symptoms such as depression, paralysis due to involvement of the peripheral nervous system, loss of appetite, loss of weight, anemia, dehydration and diarrhoea. Mortality can be substantial with virulent strains. The virus is highly immunosuppressive, thus leaving surviving birds susceptible to secondary infections [15,16]. This is caused by an early cytolytic phase in lymphoid cells, which is followed by a latent period when virus infects T-cells. This can then become transformative, with the emergence of T-cell lymphomas [15].

Few studies have been reported on sexual dimorphism in response to MDV. Though Bettridge et al. [17] reported that “MDV was also correlated more with males and/or birds with heavier weights”, gender and weight were possibly confounded in that particular study. Contrarily, a few studies have reported higher susceptibility of females [18,19,20]. Sexual dimorphism was also reported in response to vaccination against MD [21].

Different factors have been shown to contribute independently to the immune response’s sex-based disparity. Hormonal mediators such as testosterone and estrogen are reported to be immunosuppressive substances and modulators of differentiation, maturation and lifespan in various innate immune cell lineages [22], with fundamental differences in male and female life histories [23]. However, higher susceptibility to infections was observed in heterogamete gender from birth to adulthood. In accordance, it was reported that mammalian females tend to live longer than males, while male birds tend to live longer than females [24]. On average the homogametic sex lives 17.6% longer than the heterogametic sex. Intriguingly, homogametic males live only 7.1% longer than females, while homogametic females live no less than 20.9% longer than males [21]. The so called “unguarded X” hypothesis suggests that the reduced or absent chromosome in the heterogametic sex, exposes recessive deleterious mutations on the other sex chromosome. These findings suggest that sex chromosomes rather than sex hormones have a major role in sexual dimorphism of immune response.

Innate immunity is the first line of defense against pathogens, and also plays a fundamental role in the activation, regulation, and orientation of the adaptive immune response [22]. Interestingly, in mammals several genes encoding innate immune molecules are located on the X chromosome, and this may have significant differential consequences on their expression in each gender. Similarly, various genes involved in the immune response are mapped on the chicken Z chromosome (GGZ), such as the interferon (IFN) cluster, complement component genes, interleukin receptors, TRIM genes, and others [10].

Currently available vaccines prevent the formation of tumours, but do not prevent MDV infection or shedding of the pathogenic virus [25]. Consequently, both vaccine and pathogenic MDVs are found in vaccinated flocks, resulting in the emergence of increasingly more virulent strains [26]. As more virulent strains emerge, vaccine treatments are becoming less and less effective [27].

The emergence of these new and more virulent virus strains calls for additional means of control, such as genetic improvement. Indeed, genetic selection to aid breeding for viral resistance has already been applied to improved survival in commercial chicken populations [28]. In recent years, genomic selection has become widely and successfully used in plants and animals [29]. Nevertheless, it was shown that the involved truncation selection based on genomic estimated breeding values could reduce the full potential of genetic value by up to 40% in the long term, due to allele loss of favorable quantitative trait loci (QTL) [30]. Knowing the genomic elements and causative variants associated with a trait can preserve the variation and prevent that loss. This can be achieved by QTL mapping.

For decades, researchers have sought to identify the genomic elements responsible for MD resistance, with limited success. It has become clear that many genomic elements are involved in the resistance phenotype, most with relatively small effect, thus making it difficult to identify causal variants [31,32]. Nevertheless, it has long been known that the chicken MHC, located on microchromosome 16, has a major role in disease resistance, including for MD [32]. Hence, identification of non-MHC genes must be done within the context of MHC background to avoid confounding due to MHC segregation.

Previously we used an F_6_ population from a full-sib advanced intercross line to map MD QTL regions (QTLRs) on the chicken autosomes [33]. Here, we extend that study, to identify and test regions on GGZ associated with response to MD challenge.

## 2. Materials and Methods

### 2.1. Populations

All procedures carried out on the birds involved in this study were conducted in compliance with Hy-Line International Institutional Animal Care and Use Committee guidelines.

Nine populations described by Smith et al. [33] were used in the present study. These comprised 1192 females from five families of an F_6_ population from a Full Sib Advanced Intercross Line (FSAIL) used to map QTLRs affecting MD resistance, and 9077 males from eight elite commercial egg production lines used in that same study to test the QTLRs. The FSAIL was produced by inter-crossing two White Leghorn lines. The eight lines represented three different breeds; White Leghorn (WL), White Plymouth Rock (WPR) and Rhode Island Red (RIR). The F_6_ females were used to map QTLRs affecting age at death or survival following MDV challenge, by individual genotyping [33]. Males from the elite lines were used to map QTLRs affecting daughter MD mortality following MDV challenge, by Selective DNA Pooling (SDP). The same males were also used to test elements in the QTLRs as described in Smith et al. [33].

### 2.2. Mapping QTLRs Affecting Age at MD Death or Survival Using the F_6_ Population

At the F_6_ generation, 1615 females were challenged with vv+ MDV strain 686 following the protocol of Fulton et al. [28]. Following MDV challenge, F_6_ females were phenotyped for age at death or survival, and genotyped using SNPs located on GGZ by high density SNP array, as part of our previous study [33]. The phenotypes and genotypes were used to map MD QTLRs independently within each of the five families of this study, using JMP Genomics SNP-Trait association Trend test (JMP Genomics, Version 9, SAS Institute Inc., Cary, NC, USA, 1989–2019). As previously described [33], survival was taken as the censor variable; age at death as a survival trait and MHC as a class variable and fixed effect (details of the MHC genotyping will be published in the future). Following Lipkin et al. [34] and continuing in Smith et al. [33], QTLs were identified using a moving average of −LogP (mAvg) of a window of ~0.1 Mb (27 markers) with steps of 1 marker, and a critical threshold of mAvg ≥ 2.0 (*p* = 0.01); QTLR boundaries were defined by Log drop 1 [35]. As in Smith et al. [33], QTLRs within 1 Mb of one another were conservatively consolidated.

### 2.3. Mapping QTLRs Affecting Daughter MD Mortality in Eight Elite Lines by Selective DNA Pooling (SDP)

Males from eight lines were used, with each line sampled across 15 generations. Sires had known MHC genotypes, and daughter tests for MD mortality following MDV challenge [36,37]. Males were mated to multiple females as part of the routine selection process within the Hy-Line, to produce 30 half-sib female progeny per sire. Progeny females were vaccinated at 1 day of age with HVT/SB1 and at 7 days of age inoculated subcutaneously with 500 PFU of vv+ MDV (provided by Avian Disease and Oncology Lab, East Lansing MI). Mortality was recorded from 3 to 17 weeks of age (termination of experiment), as described in Fulton et al. [28]. The sire MD tolerance phenotype is the proportion of survivors among the daughters upon MD challenge.

MD values corrected for MHC genotype were obtained as follows: within each line a two-way ANOVA was used to get LS estimates of MHC effects using JMP Pro (version 15.1.0, SAS Institute Inc., Cary, NC, USA), with MD mortality as the dependent variable, and generation, MHC genotype and generation x MHC interaction as model effects. A correction factor was then calculated by subtracting the LS estimate of the target genotype from the LS estimate of the most frequent MHC genotype. Finally, the corrected factor was added to the MD value of each sire carrying the target genotype.

Forty males with high or 40 males with low corrected daughter MD mortality were selected within-generations and within-lines for the phenotypic tails of the populations. A total of 192 selected DNA pools comprised of these males were constructed from the individual DNA samples of the same males used by Smith et al. [33]. All pools were genotyped by Affymetrix 600K chicken SNP array, and markers on GGZ were used in the present analysis. Following Lipkin et al. [34], frequencies of SNP alleles were estimated based on raw intensities of alleles A and B, B% = B/(A + B). P-values of the frequency difference between tails were calculated based on empirical standard error (SE) within tails, assuming no QTL effect within tails. As for the F_6_, QTLs were then identified within lines by mAvg ≥ 2.0, and QTLR boundaries were defined by Log Drop 1. Again, as in the F_6_, QTLRs within 1 Mb of one another, were conservatively merged within a line.

### 2.4. Bioinformatics

The gene content of all QTLRs was examined to identify possible candidate genes. Genes were identified using the BioMart tool within Ensembl (v101) (https://www.ensembl.org/info/data/biomart/index.html) accessed on 1 September 2020. The QTLRs obtained were also compared to the Chicken QTL database (ChickenQTLdb: https://www.animalgenome.org/cgi-bin/QTLdb/GG/index) accessed on 31 August 2020, using the options ‘align to NCBI Chicken SNPs’ and ‘align to chicken genome’.

### 2.5. Analysis of all QTLRs by Individual Genotyping

Sequence information obtained previously for each line [38], was used to identify segregating SNPs in and around all QTLRs. Markers were chosen on the basis of their location and being equally spaced as much as possible across and around the QTLR, with no regard for genetic function. Sires from the eight lines were individually genotyped at the selected markers by Kompetitive Allele Specific PCR (KASP) [39]. Once all SNP genotyping was completed, haplotypes were identified. For some of the very long QTLRs, it was difficult to define specific haplotypes, so partial haplotypes were defined within each region.

### 2.6. Linkage Disequilibrium (LD) in QTLRs

LD r^2^ values between all possible marker pairs were obtained using the same JMP Genomics software used for the association test. Calculations were carried out within each of the eight lines. As described in Lipkin et al. [40], LD blocks were defined as a group of markers having high LD with each other (r^2^ ≥ 0.7) or moderate LD (0.15 ≥ r^2^ < 0.70). The definition was applied even if markers with low LD appeared between the markers with high or moderate LD. This definition allowed a “look over the horizon” and identification of fragmented and interdigitated blocks.

## 3. Results

### 3.1. Mapping of MD QTLRs in a Full-Sib Advanced Intercross F_6_ by Individual Genotyping

F_6_ females from five families were used to map MD QTLRs on GGZ within a family. After consolidating QTLRs within 1 Mb of one another, four QTLRs were found, but only in Family 2 (Table 1, Figure 1). Finding QTLs in only one family aligns GGZ with what has been seen in the autosomes (see Section 4. Discussion).

QTLRs F6-1, F6-2 and F6-3 overlapped seven QTLRs found by Heifetz et al. [36,37], and QTLR F6-1 also overlapped a QTLR reported by McElroy et al. [41] (Table 2). QTLR F6-4 has not been reported before.

### 3.2. MD QTLR Mapping by Selective DNA Pooling (SDP) in Eight Pure Lines

Selected DNA pools of males from eight lines were used to map MD QTLRs on GGZ. Six QTLRs were found in four of the eight lines (Table 3). The first three QTLRs overlapped each other, and all were within QTLR F_6_-1 (Table 1). As described in the Methods, the three overlapping QTLR were consolidated, to give a final list of four QTLRs (Table 3, Figure 1).

Three of the QTLRs found by pools confirmed the previous reports of Heifetz et al. [36,37] (Table 4). QTLR P-4 has not been previously reported.

### 3.3. Analysis of All Regions by Individual Genotyping

The mapped QTLRs were scanned in silico for genes residing within them (Appendix A), and were tested for association with MD response by individual genotyping of sires from the same eight lines used in the SDP (Appendix A).

A total of 113 markers and 12 haplotypes were tested (Figure 1, Appendix A). Numerous significant (0.05 ≥ *p* > 0.01) to highly significant (*p* ≤ 0.01) results were obtained (exemplified in Table 5). There was a large excess of small P-values up to 0.05 or 0.10 over the proportion expected by chance, both within and across lines for both individual markers and haplotypes (Appendix A), attesting to the presence of true QTL effects.

With the above overlap between QTLRs F_6_-1 and P-1, a total of seven QTLRs were found. Taking one significant test as a validation [33], individual genotyping confirmed the association of all QTLRs (Table 6).

### 3.4. Linkage Disequilibrium among Markers from All Regions

LD r^2^ values among and between all regions distributed from 0.0 to 1.0. Blocks were defined as a group of markers sharing LD, either with high LD (r^2^ ≥ 0.70) or moderate LD (0.15 ≤ r^2^ < 0.70). Though differing between lines due to different marker informativity, blocks were found in all regions examined (Appendix A).

Some of the blocks were interdigitated and fragmented, an observation we reported previously in a similar population [40]. Table 7 presents two fragmented interdigitated high LD blocks in Line WL4 in QTLR F_6_-2. Block 1 comprises three markers (labelled in yellow), namely the first upstream Markers 23073 and 23086 and last downstream Marker 23651. This block is split by and interdigitated with Block 2 (green), including the three Markers 23217, 23250 and 23277. Note that if the markers of Block 2 were not included in the analysis (e.g., because they were not in the marker list, were filtered out by the quality control, or were not polymorphic in this line), then one clear unambiguous Block 1 would have been identified (Table 8). Thus, though mixed, the two groups of linked markers behave as a genuine LD block.

Table 7 and Table 8 also presents the *p*-values of the Trend association tests. The accord between the LD blocks and the distribution of the *p*-values is obvious. Similar to identical *p*-values were obtained within each LD block. Block 1 was significant while Block 2 was not.

These results justify the use of such blocks along with association results and bioinformatics data, to infer location of causative elements (below and in the Appendix A).

## 4. Discussion

### 4.1. General

MD QTLRS were mapped on the chicken Z chromosome using a previously utilized F_6_ population with individual genotyping and phenotype, and DNA pools of sires from eight lines with daughter tests and selective DNA pooling.

Four QTLRs were found in the F_6_ population, in a single family out of five tested. Finding QTLRs in only one family aligns GGZ with findings by Smith et al. [33], where 10 of 19 autosomes with MD QTLRs, showed QTLRs in only one family, representing 32 of the 38 identified autosomal QTLRs. This seemingly family specific QTLRs phenomenon could be a result of this population design. The two lines that were used to produce the five families that eventually produced the F_6_ were not inbred lines. Segregation of the QTLR within the original lines would result in not all families containing identical genetic variation, thus not all families would show the same QTLR. In fact, one of the values of this particular FSAIL population was the possibility of finding various segregating QTLR.

Then again, it was found previously that many QTLs/genes are involved in the resistance phenotype, most with relatively small effect, which makes the QTLRs difficult to identify [31]. Thus, the lack of identification of QTLRs in four of the five families may indicate that indeed most QTLR effects are too small for identification in the current experimental design.

Three of the four F_6_ QTLRs overlapped QTLRs found Heifetz et al. [36,37] and by McElroy et al. [41]. QTLR F_6_-4 was not reported before. Given that this is the same population used in all three studies, the overlaps are not surprising. Thus, the present study nicely confirms previous results. The use of more dense SNPs in the present study allows for finer mapping of the QTLR compared with microsatellite markers used in the previous studies.

Four QTLR were found by DNA pools in eight elite commercial egg production lines, one of which is embedded within QTLR F_6_-1. Thus, the same region on GGZ was identified by individual genotyping in F_6_, and by SDP in three lines. Three of the QTLRs found by the pools confirmed previous reports, while one QTLR has not been previously reported (and is, in fact, the most significant QTLR).

The mapped QTLRs were tested for association with MD response by individual genotyping of sires from the same eight lines used in the SDP. Taking one significant test as a validation, all QTLRs were confirmed.

LD was calculated for all possible pairs of markers, and LD blocks were identified. Some of the LD blocks were interdigitated and fragmented as we reported previously. The blocks accorded well with the distribution of the *p*-values, justifying their use to infer location of causative elements, along with association results and bioinformatics data.

### 4.2. Detailed Analysis of Three Regions

Results of association tests, LD analysis and in silico investigation of all regions, were used to assess potential candidate genes for MD resistance and narrow possible location of causative elements. Thorough examination of the distribution of the *p*-value locations and LD pattern revealed some interesting observations, but also shows the complexity of such analysis with the limited available data. Three QTLRs are presented below, with the remainder of the QTLRs detailed in the Appendix A.

#### 4.2.1. QTLRs F_6_-1 + P-1

In this study alone, the same region was identified in no less than four independent populations, namely F_6_, Line WL2, Line WL4 and Line RIR1.

In the F_6_ + P-1 region, significant tests with similar *p*-values were obtained in Lines WL2 and WL4, where this QTLR was found (Table 3). However, QTLR P-1 was also found in Line RIR1, where no marker was significant by individual genotyping. Then again, further examination show that this line had no informative markers covering the region 7.2–7.5 Mb, initially identified in this line as a QTLR (Table 3, Appendix A). Thus, individual genotyping confirmed the pools results in Lines WL2 and WL4, while no information was available for Line RIR1.

Line WL2 presented a seemingly simple distribution of P-values with three consecutive significant Markers—70632, 71261 and 93290. This allegedly simple pattern, however, is broken up by both the QTLR mapping procedure and the LD blocks. There were no informative markers between Markers 71261 and 93290, the latter located more than 1 Mb downstream of the F_6_-1 + P-1 region. The significant Markers 70632 and 71261 in the F_6_-1 + P-1 region had perfectly equal *p* = 1.5 × 10^−2^, while Marker 93290 had a different *p* = 3.6 × 10^−2^ (Appendix A). Furthermore, while the two upstream markers were in complete LD with one another (r^2^ = 1.000), both had practically no LD with Marker 93290 (r^2^ = 0.056; Appendix A).

Thus, the combined results of association tests and LD analysis suggest two causative elements. The first most likely located within the F_6_-1 + P-1 region at about 7.0–7.2 Mb, in or near the genes *KIAA1328* (protein hinderin isoform X4) and *AQP7* (aquaporin-7 isoform X2) (Appendix A). Both genes, however, are positional candidates only, with no known relationship to MD. Thus, either one of the genes has unknown effect on the response to MD, or another yet unknown element in this region is the causative one. The second putative causal element is indicated to be more than 2 Mb downstream, in the downstream F_6_-1 only region, close to the *CCBE1* gene (collagen and calcium-binding EGF domain-containing protein 1 isoform X1). CCBE1 has been shown to act as both a tumour suppressor [42] and as an oncogenic factor [43].

#### 4.2.2. QTLR F_6_-2

QTLR F_6_-2 was confirmed by 5 significant tests in Lines WL2, WL3 and WL4, and by 2 tests across lines (Table 6 and Appendix A). These results suggest a single causative element segregating in Lines WL2 and WL3, in the region of the *PDE8B* gene (Phosphodiesterase 8B). In human this gene was found related to striatal degeneration [44] and thyroid carcinoma [45].

Line WL4 exemplifies the complexities of *p*-value distribution and LD pattern occasionally seen in GWAS and LD analyses. Three markers were significant—the adjacent Markers 23073 and 23086, and the separate Marker 23651 (Appendix A). The significant pair and the single marker were separated by three non-significant markers. The LD pattern accorded well with this fragmented distribution of the *p*-values. The upstream significant marker pair 23073-23086 form a very high fragmented LD block with the significant Marker 23651 (Table 7), located 0.56 Mb downstream. The above Marker 23217 that was significant in Lines WL3 and WL4 but not in Line WL4, formed a second block with the next two adjacent Markers 23250 and 23277, splitting and interdigitated with the previous block.

These complex results make it difficult to locate the causative elements in Line WL4 in QTLR F_6_-2. The common assumption underlying QTL mapping is that a marker is significant due to its linkage with a causative element. Based on this assumption, Block 1 is supposed to be linked to at least one causative element, while Block 2 is not. But Block 1 is split, surrounding Block 2. So where is the causative element? Clearly it is not necessarily near one of the significant markers, but could be near a distant marker in high LD with the first. Thus, the causative element may be located near Markers 23073-23086 just upstream of *PDE8B*, or near Marker 23651 in the region of the *SV2C* gene (Synaptic Vesicle Glycoprotein 2C), or there could be two causative elements—one on either side, with high LD between them.

Thus, given the fragmented and interdigitated LD blocks, while using LD to aid localization of causative elements, one must look further beyond the non-significant markers, for possible distant markers and blocks in high LD with the significant block. The causative element may, in fact, reside there.

All in all, these results indicate one causative element distributing in Lines WL2 and WL3, and a different causative element or elements distributing in Line WL4.

#### 4.2.3. QTLR P-4

QTLR P-4 was first reported in this study. This was the most significant QTLR tested by individual genotyping in the eight lines, with the highest number of significant results, the highest number of lines in which it was significant, and the highest number of significant *p*-values (Table 5). Lines WL1, WPR1, WPR2 and RIR1 had significant tests in the range *p* = 4.6 × 10^−2^ to 6.8 × 10^−9^ (note that QTLR P-4 was identified in Line WPR2), while Across Lines P-values ranged from 1.9 × 10^−3^ to 1.3 × 10^−13^.

Plotting *p*-values against location (Figure 2), Lines WPR1 and WPR2 peaked at Marker 79463 located at 79.5 Mb. The Across Lines tests peaked at 79.7 Mb, decreased at Marker 79463, but peaked in the next marker. This decrease, however, is most likely the result of a dilution effect by the three non-significant Lines WL2, WL5 and RIR1 tested here and not in the markers around it (Table 5, Figure 2). In both Lines WPR1 and WPR2, the four most significant Markers, from 78892 to 79671, formed a high LD block, and limited LD (if at all) with other markers in the QTLR (Appendix A). These results suggest one causative element toward the distal part of QTLR P-4.

The first marker in this QTLR (73225) is in the single candidate gene identified here, namely *GTF2H2* (General Transcription Factor IIH Subunit 2). GTF2H2 has been linked in human to spinal muscular atrophy [46] and to the neurodegenerative disorder Cockayne syndrome [47].

QTLR P-4 appears to be the prime region for a follow-up finer mapping study.

## 5. Conclusions

Complex LD patterns necessitate searching for fragmented interdigitated LD blocks in regions around the significant LD block, beyond the non-significant markers. The confirmed QTLRs and their candidate genes may explain at least part of the gender difference in MD response. This information can be used to help improve the chicken response to MD. The data and results presented in this study exemplify the complexity of MD resistance in chickens. Even though multiple studies have reported specific genes, or genetic regions which can have an impact on resistance to this virus, these can be line-specific or even viral strain specific. However, with specific candidate genes identified, further studies on these particular genes will likely provide additional insights into the overall function of these genes in viral immunity.

## Figures and Tables

**Figure 1 genes-14-00020-f001:**
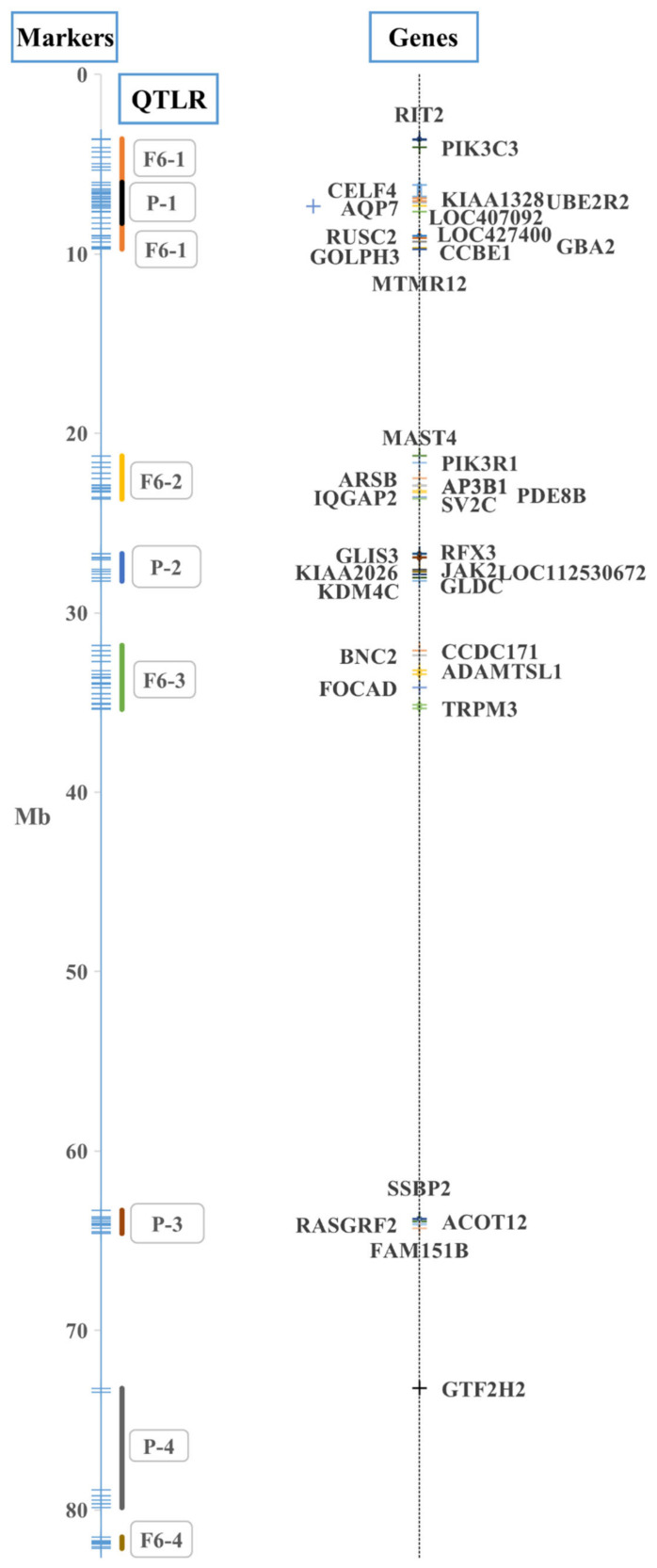
QTLR analysis. Markers (Appendix A), QTLRs (Table 1 and Table 3), and genes (Appendix A) on GGZ used in the present study. QTLRs are presented by the location of the markers used to scan them (Appendix A); as the markers also included the flanks of the regions, the boundaries in the figure are somewhat different from Table 1 and Table 3; Mb, megabase position on the GRCg6a chicken genome assembly.

**Figure 2 genes-14-00020-f002:**
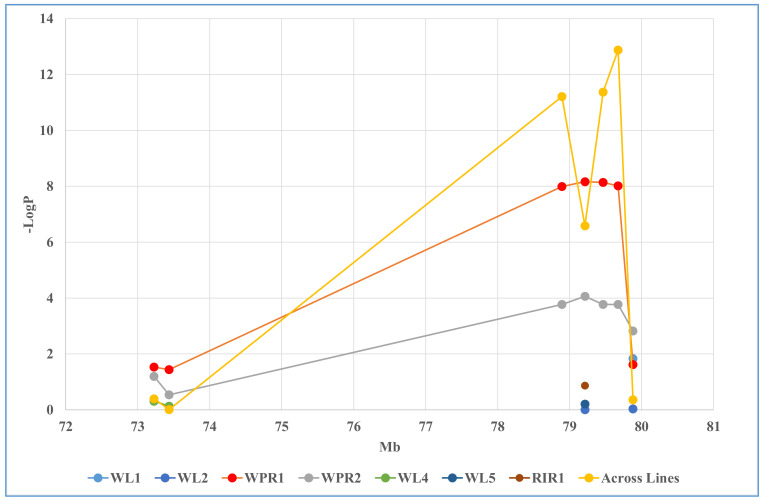
Markers -LogP plotted over locations in QTLR P-4. WL3 is not presented, as it had no informative markers within QTLR P-4.

**Table 1 genes-14-00020-t001:** QTLRs found on Chromosome Z in F_6_ Family 2.

QTLR	Start	End	Length	Distance
F_6_-1	3,436,518	9,672,022	6,235,505	
F_6_-2	21,264,317	23,659,630	2,395,314	11,592,295
F_6_-3	31,809,463	35,385,408	3,575,946	8,149,833
F_6_-4	81,712,738	81,894,873	182,136	46,327,330

QTLR, QTLR serial number within dataset; Start, End, bp location on the GRCg6a reference of the first and last markers in the QTLR; Length, the size of the QTLR in bp; Distance, bp between the start of the QTLR and the end of the previous QTLR.

**Table 2 genes-14-00020-t002:** Overlaps between QTLRs found in F_6_ Family 2 and previous reports.

F6			QTLdb	
QTLR	Start	End	Reference	Mb
F_6_-1	3,436,518	9,672,022	36	0–19.1
			37	0–36.9
			41	9.2–10.2
F_6_-2	21,264,317	23,659,630	36	20.2–28.8
			37	0–36.9
			37	13.9–27.0
F_6_-3	31,809,463	35,385,408	37	0–36.9
			37	17.1–66.9
F_6_-4	81,712,738	81,894,873		

QTLdb, ChickenQTLdb: https://www.animalgenome.org/cgi-bin/QTLdb/GG/index accessed on 31 August 2020.

**Table 3 genes-14-00020-t003:** QTLRs found on Chromosome Z by pools of four of the eight pure lines.

Analysis	Line	QTLR	Start	End	Length	Distance
By Line	WL2	1	6,802,425	7,250,635	448,211	
	WL4	2	7,138,346	7,297,108	158,763	−112,289
	RIR1	3	7,208,372	7,459,200	250,829	−88,736
	WPR2	4	27,496,986	27,807,851	310,866	20,037,786
	WL2	5	63,831,711	64,162,739	331,029	36,023,860
	WPR2	6	78,740,304	79,239,183	498,880	14,577,565
Consolidated across lines	WL2, WL4, RIR1	P-1	6,802,425	7,459,200	656,776	
	WPR2	P-2	27,496,986	27,807,851	310,866	20,037,786
	WL2	P-3	63,831,711	64,162,739	331,029	36,023,860
	WPR2	P-4	78,740,304	79,239,183	498,880	14,577,565

Analysis: By line, QTLR mapping within line; Consolidated across lines, QTLRs within 1 Mb of each other were consolidated across lines. QTLR, QTLR serial number within dataset; Start, End, bp location on the GRCg6a reference of the first and last markers in the QTLR; Length, the size of the QTLR in bp; Distance, bp between the start of the QTLR and the end of the previous QTLR; negative distance indicates overlap between QTLRs.

**Table 4 genes-14-00020-t004:** Overlaps between QTLRs found by the pools of the eight pure lines and previous reports.

Pools				QTLdb	
Lines	QTLR	Start	End	Reference	Mb
WL2, WL4, RIR1	P-1	6,802,425	7,459,200	37	0–36.9
WPR2	P-2	27,496,986	27,807,851	37	0–36.9
				36	20.2–28.8
				37	13.9–27.0
				37	17.1–66.9
WL2	P-3	63,831,711	64,162,739		
WPR2	P-4	78,740,304	79,239,183		

Pools, QTLRs from Table 3; QTLdb, ChickenQTLdb: https://www.animalgenome.org/cgi-bin/QTLdb/GG/index accessed on 31 August 2020.

**Table 5 genes-14-00020-t005:** An example of association tests by individual genotyping, of all markers and haplotype in QTLR P-4 (Appendix A).

QTLR	Marker/Haps	bp	Distance	Line								AcrossLines	
				WL1	WL2	WL3	WPR1	WPR2	WL4	WL5	RIR1		Gene
P-4	73225	73,225,511	8,641,932				2.9 × 10^−2^	6.3 × 10^−2^	4.9 × 10^−1^			4.0 × 10^−1^	GTF2H2
P-4	73435	73,435,755	210,244				3.6 × 10^−2^	2.9 × 10^−1^	7.4 × 10^−1^			9.8 × 10^−1^	
P-4	78892	78,892,243	5,456,488				1.0 × 10^−8^	1.7 × 10^−4^				6.1 × 10^−2^	
P-4	79212	79,212,166	319,923		9.9 × 10^−1^		6.8 × 10^−9^	8.7 × 10^−5^		6.2 × 10^−1^	1.4 × 10^−1^	2.6 × 10^−7^	
P-4	79463	79,463,296	251,130				7.2 × 10^−9^	1.7 × 10^−4^				4.2 × 10^−2^	
P-4	79671	79,671,839	208,543				9.6 × 10^−9^	1.7 × 10^−4^				1.3 × 10^−3^	
P-4	79878	79,878,110	206,271	1. × 10^−2^	9.2 × 10^−1^		2.4E × 10^−2^	1.5 × 10^−3^				4.4 × 10^−1^	
P-4	Haps								7.2 × 10^−1^	6.3 × 10^−1^	4.6 × 10^−2^	1.9 × 10^−3^	

QTLR, QTLR serial number found by the Pools (Table 3); Marker/Haps, marker or haplotype tested; bp, location on GGZ (haplotypes have no specific location); Distance, bp between markers; Line, test within a line; Across lines, test across all lines; pink highlight, *p* ≤ 0.05; Gene, a gene found in the QTLR (Appendix A). Markers are ordered by location (GRCg6a).

**Table 6 genes-14-00020-t006:** Examination of the QTLRs by individual genotyping.

QTLR	Marker Tests				Haplotype Tests				Sum		Confirmed
	Markers		Ac Lines		Lines		Ac Lines				
	Tests	Sig	Tests	Sig	Tests	Sig	Tests	Sig	Tests	Sig	
F_6_-1, P-1	50	4	26	5	6	2	0	-	82	11	✓
F_6_-2	45	5	16	2	6	0	2	0	69	7	✓
P-2	13	0	7	3	0	-	1	1	21	4	✓
F_6_-3	24	1	13	0	2	0	2	0	41	1	✓
P-3	20	3	8	0	0	-	1	0	29	3	✓
P-4	21	13	7	4	3	1	1	1	32	19	✓
F_6_-4	23	4	7	1	2	0	0	-	32	5	✓

QTLR, QTLR serial number as found by F_6_ (Table 1) and the Pools (Table 3); Ac Lines, across lines; Tests, number of tests conducted; Sig, number of test significant at *p* ≤ 0.05.

**Table 7 genes-14-00020-t007:** Fragmented interdigitated blocks found in Line WL4 in QTLR F_6_-2. All markers. LD Block 1 (yellow) is fragmented and interdigitated with Block 2 (green).

Gene,			PDE8B	PDE8B	PDE8B	SV2C
Bp,	23,073,692	23,086,227	23,217,357	23,250,769	23,277,739	23,651,225
Dis.,		12,535	131,130	33,412	26,970	373,486
Marker,	23073	23086	23217	23250	23277	23651
23073						
23086	1.000					
23217	0.088	0.088				
23250	0.090	0.090	0.981			
23277	0.090	0.090	0.981	0.990		
23651	0.830	0.830	0.082	0.084	0.084	
*p*:	2.82 × 10^−2^	2.82 × 10^−2^	5.10 × 10^−1^	5.12 × 10^−1^	4.88 × 10^−1^	1.61 × 10^−3^

Markers are ordered by location. Mb, location on GRCg6a in Mb; Dis, distance in bp from the pre-vious marker; Marker, number of the marker; yellow and green, LD Blocks 1 and 2; red, LD r^2^ ≥ 0.7; white, r^2^ < 0.15; p, p-value of the Trend association test (Appendix A): pink highlight, p ≤ 0.05; white, *p* > 0.05.

**Table 8 genes-14-00020-t008:** Fragmented interdigitated blocks found in Line WL4 in QTLR F_6_-2. LD Block 1 alone. Without the markers of Block 2 (Table 7), one clear unambiguous Block 1 would have been identified.

Gene:			SV2C
bp:	23,073,692	23,086,227	23,651,225
Dis.:		12,535	564,998
Marker:	23073	23086	23651
23073			
23086	1.000		
23651	0.830	0.830	
*p*:	2.82 × 10^−2^	2.82 × 10^−2^	1.61 × 10^−3^

Markers are ordered by location. Mb, location on GRCg6a in Mb; Dis, distance in bp from the previous marker; Marker, number of the marker; yellow, LD Blocks 1; red, LD r^2^ ≥ 0.7; *p*, *p*-value of the Trend association test (Appendix A): pink highlight, *p* ≤ 0.05.

## Data Availability

Not applicable.

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
