# Peer review of "Sex Differences in Response to Marek’s Disease: Mapping Quantitative Trait Loci Regions (QTLRs) to the Z Chromosome"

_genes, 2022, doi:10.3390/genes14010020_

Round 1

Reviewer 1 Report

The authors used females from Hy-line F6 population and males from eight elite commercial egg lines to map MD QTLs in GGZ. The results identified candidate MD QTLRs as well as genes in them and explain the gender differences in MD tolerance to some extent.

The suggestion and comments are as follows:

1.      I know the samples used in this study might be the same with their previous study. But if the authors give the number of females and males in this study that will help readers understand better.

2.      How selective DNA pooling (SDP) conduct? How many DNA pooled together?

3.      I think the authors should give some necessary descriptions of MHC genotypes for female or male populations.

4.      In tableS2, “bp” is location. Why not use “location” directly?

5.      In line7 of Appendix1 file, p values of three significant markers were 6.2-1.6E-3, not “3.2-1.6E-3”.

6.      In line 8, Markers 26696 and 26716 should be 22696 and 26719.

7.      In TableS1, it will be better to add the ID of QTLRs. There are some repetitive records, for example, F6-1~RIK3C3 (Line 4-8 in Table S1 ). Why?

Author Response

Many thanks for the reviewe. our red replies to your comments are:

  1. I know the samples used in this study might be the same with their previous study. But if the authors give the number of females and males in this study that will help readers understand better. 1,192 was added to line 125 for F6 females and 9,077 to line 126 for males from the 8 lines.
  2. How selective DNA pooling (SDP) conduct? How many DNA pooled together? SDP is a well-documented method. The pooling procedure is detailed in lines 149-177 with references. "Forty" and "40 males with" were added to line 157. "A total of 192" and "comprised of these males" were added to line 169.
  3. I think the authors should give some necessary descriptions of MHC genotypes for female or male populations. "(details of the MHC genotyping will be published in the future)" was added to lines 143-144.
  4. In tableS2, “bp” is location. Why not use “location” directly? "bp" was changed to "Location".
  5. In line7 of Appendix1 file, p values of three significant markers were 6.2-1.6E-3, not “3.2-1.6E-3”. "3.2" was corrected to "6.2" in line 44.
  6. In line 8, Markers 26696 and 26716 should be 22696 and 26719. 26716 was corrected to 26719 in line 45.
  7. In TableS1, it will be better to add the ID of QTLRs. The first column in the table is the QTLR. There are some repetitive records, for example, F6-1~RIK3C3 (Line 4-8 in Table S1 ). Why? All repetitive records were removed.

Reviewer 2 Report

The research used a full-sib advanced intercross to identify regions in GGZ in response to a MD challenge.  It builds on prior studies with this AIC. 

1. Abstract - Although obvious to geneticists, should LD be defined?  You explain it (l. 187-193).

2. The reference section requires editing.  Hopefully it does not reflect on attention to detail in the experiment.

3. If provided in this submission, I missed the MD challenge description.

4. What was the average (or range) in size of the sire families?  Was it consistent across generations?

5. l. 127. Is a WPR a typical commercial egg producing line?

6. l. 379. Are you comfortable with "insignificant markers"?  The implication is that you found markers that you felt were not relevant, rather than relying on probabilities.

7. l. 416. "perhaps utilizing gene editing tools"  I am puzzled why this phrase is a specific conclusion from your study.

Author Response

Many thanks for the reviewe. our red replies to your comments are:

  1. Abstract - Although obvious to geneticists, should LD be defined?  You explain it (l. 187-193). LD was changed to "Linkage Disequilibrium (LD)" in lines 25-26.
  2. The reference section requires editing.  Hopefully it does not reflect on attention to detail in the experiment. Edited.
  3. If provided in this submission, I missed the MD challenge description. "At the F6 generation, 1,615 females were challenged with vv+ MDV strain 686 following the protocol of Fulton et al. [28]" was added to lines 136-137. "Males were mated to multiple females as part of the routine selection process within the Hy-Line, to produce 30 half-sib female progeny per sire. Progeny females were vaccinated at 1 day of age with HVT/SB1 and at 7 days of age inoculated subcutaneously with 500 PFU of vv+ MDV (provided by Avian Disease and Oncology Lab, East Lansing MI). Mortality was recorded from 3 to 17 weeks of age (termination of experiment), as described in Fulton et al. [28]. The sire MD tolerance phenotype is the proportion of survivors among the daughters upon MD challenge" was added to lines 153-159.
  4. What was the average (or range) in size of the sire families?  Was it consistent across generations? As is described in now line 154, "30 half sib female progeny per sire".
  5. l. 129. Is a WPR a typical commercial egg producing line? The RIR and WPR are the breeds used to produce the commercial brown egg laying varieties.
  6. l. 379. Are you comfortable with "insignificant markers"?  The implication is that you found markers that you felt were not relevant, rather than relying on probabilities. Changed "in" to non-" in line 389.
  7. l. 416. "perhaps utilizing gene editing tools"  I am puzzled why this phrase is a specific conclusion from your study. These words were removed from now line 427.